# ELISA-Based Assay for Studying Major and Minor Group Rhinovirus–Receptor Interactions

**DOI:** 10.3390/vaccines8020315

**Published:** 2020-06-18

**Authors:** Petra Pazderova, Eva E. Waltl, Verena Niederberger-Leppin, Sabine Flicker, Rudolf Valenta, Katarzyna Niespodziana

**Affiliations:** 1Department of Pathophysiology and Allergy Research, Center for Pathophysiology, Infectiology and Immunology, Medical University of Vienna, Währinger Gürtel 18-20, A-1090 Vienna, Austria; petra.pazderova@meduniwien.ac.at (P.P.); sabine.flicker@meduniwien.ac.at (S.F.); rudolf.valenta@meduniwien.ac.at (R.V.); 2Department of Otorhinolaryngology, Medical University of Vienna, Währinger Gürtel 18-20, A-1090 Vienna, Austria; eva.waltl@meduniwien.ac.at (E.E.W.); verena.niederberger@meduniwien.ac.at (V.N.-L.); 3NRC Institute of Immunology FMBA of Russia, 115478 Moscow, Russia; 4Laboratory for Immunopathology, Department of Clinical Immunology and Allergy, Sechenov First Moscow State Medical University, 119435 Moscow, Russia; 5Karl Landsteiner University of Health Sciences, 3500 Krems, Austria

**Keywords:** rhinovirus, asthma, virus receptor, virus receptor interaction assay, virus neutralization assay, intercellular adhesion molecule 1, low density lipoprotein receptor, vaccine development

## Abstract

Rhinovirus (RV) infections are a major cause of recurrent common colds and trigger severe exacerbations of chronic respiratory diseases. Major challenges for the development of vaccines for RV include the virus occurring in the form of approximately 160 different serotypes, using different receptors, and the need for preclinical models for the screening of vaccine candidates and antiviral compounds. We report the establishment and characterization of an ELISA-based assay for studying major and minor group RV–receptor interactions. This assay is based on the interaction of purified virus with plate-bound human receptor proteins, intercellular adhesion molecule 1 (ICAM-1), and low density lipoprotein receptor (LDLR). Using RV strain-specific antibodies, we demonstrate the specific binding of a panel of major and minor RV group types including RV-A and RV-B strains to ICAM-1 and LDLR, respectively. We show that the RV–receptor interaction can be blocked with receptor-specific antibodies as well as with soluble receptors and neutralizing RV-specific antibodies. The assay is more sensitive than a cell culture-based virus neutralization test. The ELISA assay will therefore be useful for the preclinical evaluation for preventive and therapeutic strategies targeting the RV–receptor interaction, such as vaccines, antibodies, and anti-viral compounds.

## 1. Introduction

Rhinovirus (RV) infections are the major cause of recurrent common colds and are responsible for more than 80% of wheeze and asthma exacerbations in children [1,2]. RV-induced wheeze in early childhood seems to predispose children to developing recurrent wheeze and asthma later in life [3].

Besides allergic sensitization, RV infections are among the most common triggers for asthma [4]. More than 160 different RV types are known [5]. They can be divided into three RV species—RV-A, RV-B, and RV-C—according to their primary sequence; they are known to bind to defined receptors on their host cells. The majority of major group RVs belonging to RV-A and RV-B species bind to intercellular adhesion molecule 1 (ICAM-1) [6], and a subset of RV-A species belonging to minor group RVs target the low density lipoprotein receptor (LDLR) [7]. The more recently described RV-C species, which, according to sequence homology, are more closely related to RV-A than to RV-B species, has been reported to use cadherin-related family member 3 (CDHR3) protein as the receptor [8]. Whereas RV-A and RV-B species can be cultivated easily, RV-C species are difficult to propagate [9] and there are currently no infection models available to study their relevance in the etiology of asthma exacerbations.

Several lines of evidence confirm the importance of RV infections as triggers for exacerbations of severe respiratory diseases such as childhood wheeze, asthma, and chronic obstructive pulmonary disease. Firstly, the association of RV infection and respiratory disease exacerbations was demonstrated by the presence of virus in the respiratory tract of patients experiencing the exacerbations by showing the presence of virus-derived nucleic acids [10]. Experimental infection models showed that controlled infection with RV can trigger asthma attacks [11]. In particular, subjects suffering from allergic sensitizations seem to be especially susceptible to RV-triggered asthma [12,13], which may be related to their Th-2-biased immune system being less able to combat RV infections [14]. The finding that the majority of RV-specific antibodies, which develop in the course of RV infections, are directed against an N-terminal peptide of the VP1 capsid protein [15], allows the dissecting of RV-species-specific antibody responses and relating of the increases in RV species-specific antibody responses to respiratory disease exacerbations [16,17,18]. The cumulative RV-specific immune responses, which seem to be associated with respiratory illness, could be estimated with the micro-arrayed VP1 peptides [19].

Despite substantial progress in the species-specific detection and diagnosis of RV infections by nucleic acid-based testing and antibody testing, there is still no vaccine or RV-specific treatment available, which could be used for the prevention or treatment of RV-triggered respiratory illness. The development of RV-specific treatments and vaccines is challenging due to the large variety of RV types that use different receptors. A systematic development of vaccines, therapeutic antibodies, and compounds that inhibit the infection of the host cell by the virus requires the availability of preclinical models that allow the fast and high-throughput screening of candidate molecules. For RV, only cumbersome and crude cell-based neutralization assays [20] are currently available as well as mouse models that are based on transgenic animals expressing human receptors (e.g., ICAM-1 knock-in mice) [21]. 

To provide a robust and simple model for studying the interaction of the majority of major and minor group RVs, and thus, RV-A and RV-B species, we developed an ELISA-based interaction assay using purified RVs and the corresponding human receptor proteins. We report the establishment and characterization of these assays and demonstrate their utility for identifying antibodies and compounds that block the virus–receptor interaction.

## 2. Materials and Methods

### 2.1. Propagation of RV Strains

The H1 HeLa cell line (American Type Culture Collection (ATCC), Manassas, VA, USA) was cultured in Minimum Essential Medium (MEM) containing Earle’s salt (Gibco, Thermo Fisher Scientific, Waltham, MA, USA), supplemented with 10% fetal bovine serum (FBS; Gibco, Thermo Fisher Scientific, Waltham, MA, USA), antibiotics (penicillin–streptomycin (10,000 U/mL), and 60 µL gentamycin (1.2 µg/mL, Gibco, Thermo Fisher Scientific, Waltham, MA, USA). Cells were grown in a Cytoprem 2 CO_2_ incubator (Thermo Fisher Scientific, Waltham, MA, USA) at 37 °C, 5% CO_2_, and 91% relative humidity. Human rhinovirus strains (major group RV-A—RV89, RV8, RV16, RV18, RV54; minor group RV-A—RV2, RV25, RV29, RV44; major group RV-B—RV14, RV72) obtained from ATCC were added to MEM supplemented with 1% FBS, 15 mM magnesium chloride (Sigma-Aldrich, Merck KGaA, Darmstadt, Germany) without antibiotics (infectious medium), and used to infect HeLa cells that had been grown in a 25 cm^2^ cell culture flask with a canted neck and screwed cap (TPP Techno Plastic Products AG, Trasadingen, Switzerland) until they had reached 70–80% confluency. Then, cells were incubated for 2 h at 34 °C, 5% CO_2_, and 91% relative humidity; washed with Dulbecco’s phosphate-buffered saline (DPBS; Gibco, Thermo Fisher Scientific, Waltham, MA, USA); incubated in infectious medium for the following 1–7 days, depending on the strain being propagated. When the cytopathic effect (CPE) was observed, dead cells were centrifuged 5 min at 1200 rpm (210× *g*; Rotanta 46RC, Hettich, Tuttlingen, Germany). The supernatant was used for the infection of HeLa cells that had been grown to confluency in a 150 cm^2^ cell culture flask with a canted neck and vented cap (Corning, New York, NY, USA), following the same procedure. Finally, HeLa cells that had been grown to confluence in 10 cell culture flasks 150 cm^2^ in size were infected to obtain sufficient amounts of virus.

### 2.2. Purification of RV Strains

Dead cells were centrifuged for 20 min at 1200 rpm (210× *g*; Rotanta 46RC, Hettich, Tuttlingen, Germany). This first supernatant was collected and the pellet was suspended in 10 mL cell lysis buffer (pH 7.4) containing 0.38% (*w*/*v*) ethylenediaminetetraacetic acid (EDTA; BIOMOL, Hamburg, Germany) and 0.12% (*w*/*v*) NaCl (Merck-Millipore, Merck, Darmstadt, Germany). The suspension was subjected to three freeze–thaw cycles performed in liquid nitrogen and cold water, respectively, and further homogenized using a Dounce Homogenizer (Carl Roth, Karlsruhe, Germany). This homogenate was centrifuged for 20 min at 4000 rpm (2330× *g*, Rotanta 46RC, Hettich, Tuttlingen, Germany). The second supernatant was then combined with the first, and Polyethylenglycol 6000 (PEG_6000_, 7% *w*/*v*; Merck-Millipore, Merck, Darmstadt, Germany) and NaCl (3% *w*/*v*) were added. The virus-containing solution was incubated by moderate shaking overnight at 4 °C and was centrifuged for 20 min at 4 °C, 4000 rpm (2330× *g*, Rotanta 46RC, Hettich, Tuttlingen, Germany). The viral pellets were suspended in 10 mL DPBS and by vortexing for 1 h at room temperature. The supernatant containing purified virus preparations, after the final centrifugation for 5 min at 4 °C, 12,700 rpm (18,213× *g*, Eppendorf 5427R, Hamburg, Germany), was aliquoted and stored at −80 °C (200 µL/stock). The protein contents of purified virus preparations were determined using the bicinchoninic acid assay (BCA) Protein Assay Kit (Thermo Fisher Scientific, Waltham, MA, USA).

### 2.3. Analysis of Virus Preparations

Viral RNA was extracted using a QIAmp Viral RNA Kit (Qiagen GmbH, Hilden, Germany). Viral RNA was transcribed into cDNA and amplified by one-step reverse transcription (RT)-PCR using SuperScript III First Strand Synthesis System SuperMix (Invitrogen, Carlsbad, CA, USA) with VP4/VP2 forward (5′-GGG ACC TAC TTT GGG TGT CCG TGT-3′) and reversed (5′-GCA TCI GGY ARY TTC CAC CAC CAN CC-3′) primers (Eurofins Genomics, Ebersberg, Hamburg, Germany), where I = inosine; Y = C, T; R = A, G; N = A, G, C, T. RT-PCR was conducted in a thermal cycler MWG Primus 96 plus PCR (Stratagene, Minden, Germany) by performing reverse transcription for 1 h at 50 °C followed by 20 min of denaturation at 94 °C and 40 cycles of amplification each comprising 1 min denaturation at 94 °C, 1 min annealing at 55 °C, and 1 min elongation at 72 °C. A final extension was performed for 10 min at 72 °C. The amplicons in a total volume of 50 µL were mixed with 6× DNA Loading Dye (Thermo Fisher Scientific, Waltham, MA, USA) and subjected to 1% agarose gel electrophoresis (Biozym LE Agarose, Biozym Scientific GmbH, Hessisch Oldendorf, Germany) using ethidium bromide for staining of DNA (ethidium bromide solution; Sigma-Aldrich, Merck, Darmstadt, Germany). A DNA Ladder GeneRuler 1 kb Plus DNA Ladder (Thermo Fisher Scientific, Waltham, MA, USA) was used as the marker. PCR products were extracted from the gel and purified with Wizard^®^ SV Gel and PCR Clean-Up System (Promega Corporation, Madison, WI, USA) and used for cycle sequencing on an ABI 3730XL sequencer with the same forward and reversed primers that had been used for amplification (Eurofins Genomics, Ebersberg, Hamburg, Germany).

Forward and reversed sequences were aligned using the BLASTn program (https://blast.ncbi.nlm.nih.gov/Blast/) to identify unambiguous double strand sequences, which were compared with the sequences deposited in the database of the National Center for Biotechnology Information (NCBI) using web search engine BLASTn. These sequences were translated into protein sequences using the ExPASy translation tool (https://www.expasy.org/) and subjected to multiple sequence alignment by Clustalθ with ClustalW as an output format (https://www.ebi.ac.uk/Tools/msa/clustalo/). Editing was performed with GeneDoc 2.7 software (available at https://genedoc.software.informer.com/2.7/). Conserved regions along with conservative, semi-conservative, and non-conservative exchanges were visualized in Microsoft Word 2010 (Microsoft Corporation, Redmond, WD, USA) and Adobe Illustrator CS (64 bit; Adobe, San Jose, CA, USA).

Evolutionary analysis of protein sequences was performed using the maximum likelihood (ML) method implemented in the Molecular Evolutionary Genetics Analysis (MEGA) software version X (64 bit; available at https://www.megasoftware.net). As a statistical method, the bootstrap method with 1000 repeats was applied [22]. Before the generation of phylogenetic tree, a substitution model with the lowest Bayesian information criterion (BIC) and Akaike information criterion corrected (AICc) was estimated by the Find Best Protein Models tool implemented in MEGA-X. From the result, the general matrix (LG) model was chosen [23]. The tree was further rearranged with the nearest neighbor interchange (NNI) option [24]. 

A phylogenetic tree was constructed from the 11 representative major group and minor group RV strains. All positions with less than 95% site coverage were eliminated from the analysis (i.e., fewer than 5% alignment gaps were allowed at any position; partial deletion option). 

### 2.4. Analysis of Recombinant Human Receptor Proteins

Recombinant human ICAM-1/CD54 and LDLR were obtained from R&D Systems (Minneapolis, MN, USA). The purity and molecular weight of ICAM-1 and LDLR were determined by SDS-PAGE gel under reducing conditions and subsequent Coomassie-blue staining of the gel [25]. A protein ladder, PageRulerTM (Thermo Fisher Scientific, Waltham, MA, USA), was used as the molecular weight marker. The secondary structure of the proteins was studied by circular dichroism (CD) spectroscopy in the far ultraviolet (UV) wavelength region between 260 and 190 nm using a Jasco J-810 spectropolarimeter (Japan Spectroscopic Co., Tokyo, Japan). Samples dissolved at a concentration of 0.1 mg/mL in PBS were analyzed in 0.2 cm path length cuvettes (Hellma Analytics, Munich, Germany) at 20 °C, as previously described [26]. 

An immunological characterization of ICAM-1 and LDLR was performed by testing the proteins for reactivity with rabbit anti-ICAM-1 antibodies (Sino Biological, Beijing, China) and goat anti-LDLR antibodies from R&D Systems (Minneapolis, MN, USA). Proteins (2 µg/mL) were coated on 96-well half area microplates (Greiner Bio-One International GmbH, Kremsmünster, Austria) overnight at 4 °C, washed twice with phosphate-buffered saline containing 0.05% (v/v) Tween 20 (PBS-T), and blocked with 2% (*w*/*v*) bovine serum albumin (BSA) (Roth, Karlsruhe, Germany). Bound rabbit antibodies were detected with 1:2000 diluted horseradish peroxidase (HRP)-labeled donkey anti-rabbit antibodies (Amersham, Marlborough, MA, USA), and bound goat antibodies were detected with rabbit HRP-labeled anti-goat antibodies (R&D Systems, Minneapolis, MN, USA).

### 2.5. Receptor- and Virus-Specific Antibodies

Specific rabbit antibodies were obtained by immunizing rabbits with 200 µg aliquots of recombinant ICAM-1 (R&D Systems, Minneapolis, MN, USA) and purified recombinant VP2, VP3, and VP4 that had been purified as described [15]. The first immunizations were conducted with a mix of the respective protein and Freund’s Complete Adjuvant (CFA) (Charles River, Chatillon sur Chalaronnne, France) and the 2nd to 4th immunization with a mix of each protein and Freund’s Incomplete Adjuvant (IFA) (Charles River, Chatillon sur Chalaronnne, France). Rabbit anti-VP1 antibodies were as previously described [27]. Commercial rabbit anti-ICAM-1 (Sino Biological, Beijing, China) and goat anti-LDLR antibodies (R&D Systems, Minneapolis, MN, USA) used in certain experiments were purchased. Normal rabbit antibodies used for control experiments were obtained from Genscript (Leiden, Netherlands). RV-strain-specific guinea pig antibodies were purchased from ATCC. 

### 2.6. ELISA-Based Virus–Receptor Interaction Assay

Half-area 96-well microplates (Greiner Bio-One International GmbH, Kremsmünster, Austria) were coated with 2 µg/mL of ICAM-1, LDLR, and human serum albumin (HSA; negative control) (Sigma-Aldrich, St. Louis, MO, USA) overnight at 4 °C as described [18]. Plates were washed twice with phosphate-buffered saline containing 0.05% (v/v) Tween 20 (PBS-T), blocked with 2% (*w*/*v*) bovine serum albumin (BSA), and incubated for 4 h at 24 °C. Plates were then incubated with major group or minor group RV strains (50 µg/mL, according to the protein determination of the viral stock) overnight at 4 °C. Receptor-bound virus was then detected with strain-specific guinea pig antibodies (ATCC; 1:5000) followed by HRP-labeled sheep anti-guinea pig antibodies (Jackson IR, Cambridgeshire, UK). The color reaction was developed with substrate solution containing ABTS and H_2_O_2_, as previously described [18]. The mean optical density (O.D.) values corresponding to the amount of receptor-bound virus were measured at 405 and 492 nm (reference) in a TECAN Infinite F50 ELISA reader with the integrated software i-control 2.0 (Tecan Group Ltd., Männedorf, Switzerland). Measurements were recorded in duplicate wells and are reported as mean values. The deviations of the duplicate determinations were always less than 10% and are indicated in each of the figure legends.

### 2.7. ELISA-Based Inhibition of Virus–Receptor Interaction by Anti-Receptor Antibodies, Recombinant Receptors, or RV-Specific Antibodies

In all inhibition assays, ELISA microplates were coated with recombinant human ICAM-1 and LDLR (2 µg/mL), washed, and blocked, as described in Section 2.6. In blocking experiments performed with the rabbit-anti-ICAM-1 antibodies, rabbit sera, or for control purposes, sera from normal rabbits were added in different dilutions (1:10–1:100,000). Commercial anti-ICAM-1 and anti-LDLR antibodies were diluted to a concentration of 10 µg/mL for blocking experiments. After washing the plates as described in Section 2.6, virus preparations were allowed to bind and were detected. 

In fluid phase competition assays, RV89 (25 µg/mL) and RV2 (2 µg/mL) were pre-incubated in a solution with their corresponding receptors (50 µg/mL) overnight at 4 °C before they were allowed to bind to plate-bound receptors. 

The effects of antibodies specific for RV89-derived coat proteins were also studied by fluid phase inhibition. For this purpose, RV89 (25 µg/mL) was pre-incubated with each anti-VP antiserum diluted 1:5 or, for control purposes, only with buffer containing 0.5% BSA before virus was added to plate-bound receptors. The bound RV strains in the fluid phase inhibitions were detected as described in Section 2.6. The percentages inhibitions of receptor-bound virus were calculated as follows: Percentage inhibition of RV binding = 100 − O.D.^i^/O.D.^n^ × 100, where O.D.^i^ and O.D.^n^ represent the mean extinctions obtained with inhibitors versus without inhibitor, respectively. All figures were generated in GraphPad Prism 6 (GraphPad Software, San Diego, CA, USA) and Adobe Illustrator CS (64 bit; Adobe, San Jose, CA, USA).

### 2.8. Cell Culture-Based Virus Neutralization Assay

HeLa cells were plated in 96-well tissue culture plates (Corning, New York, NY, USA) at a density of 1.3 × 10^4^ cells/well and incubated overnight at 37 °C. On the following day, cells were incubated with 1:100 to 1:100,000 diluted rabbit anti-ICAM-1 antibodies or normal rabbit antibodies that had been heat-inactivated for 30 min at 56 °C, for 3 h at 37 °C in triplicate cultures. Then, antibody-containing medium was replaced by 100 Tissue Culture Infection Dose 50 (TCID_50_)/cell of RV89 and plates were incubated for the following 3 days at 34 °C. To assess the protective effect of anti-ICAM-1 blocking antibodies, cells were stained with 0.1% (*w*/*v*) crystal violet solution (Sigma-Aldrich, St. Louis, MO, USA) for 10 min and dried for 1 h at 56 °C on a hot plate. Crystal violet, in dried wells, was subsequently dissolved in 0.1% (*w*/*v*) SDS buffer (AppliChem, Darmstadt, Germany) for 30 min. The mean O.D. values ± SDs corresponding to the absorbance of dissolved crystal violet dye in each well were measured at 560 nm using in the ELISA reader.

## 3. Results

### 3.1. Development of an ELISA-Based RV–Receptor Interaction Assay

The aim of our study was to build a simple ELISA-based assay that allows investigation of the interaction of intact RV strains with their corresponding receptors using defined components (Figure 1). To achieve this goal, we established viral stocks from a panel of major group RV-A strains (RV89, RV8, RV16, RV18, and RV54) and major group RV-B strains (RV14 and RV72) that use ICAM-1 as a receptor, as well as minor group RV-A strains (RV2, RV25, RV29, and RV44) that bind to LDLR (Appendix A). These strains were propagated in HeLa cells and their identity was confirmed by RT-PCR using primers specific for the VP2–VP4 region. Appendix A shows the obtained PCR products, which were sequenced and identified by searching the nucleotide BLAST database available on the National Center for Biotechnology Information page (http://www.ncbi.nlm.nih.gov/BLAST) (Appendix A, Appendix A). The RV strains were representative of major and minor group RV-A and major group RV-B strains, as shown in the cladogram in Appendix A. 

The ELISA-based assay was constructed to allow purified virus to interact with ELISA plate-bound receptor molecules. For this purpose, we obtained purified ICAM-1 and LDLR molecules that had been expressed in cell lines and appeared as pure proteins in Coomassie-stained SDS-PAGE (Figure 2a). Recombinant glycosylated ICAM-1 migrated as a band of approximately 80 kDa and LDLR appeared as a band of 120–140 kDa (Figure 2a). Both recombinant receptors were then analyzed regarding secondary structure using circular dichroism (CD) spectroscopy. The spectrum of ICAM-1 was characterized by a maximum at 200 nm and two minima at 216 and 219 nm, indicating that the protein is folded and assumes *β*-sheet secondary structure typical for immunoglobulin (Ig)-like domains (Figure 2b). The result from the secondary structure fitting procedure estimated 46% *β*-sheets, 3% α-helices, 20% turns, and 29% unordered structure. Recombinant LDLR seems to be mainly unfolded, characterized by a maximum at 193 nm and two minima, the local at 199 nm and the global one at 203 nm (Figure 2b). Secondary structure fitting estimated a limited amount of α-helices (11%) and *β*-sheets (39%).

Next, we performed an immunological characterization of the receptor proteins. Figure 2c shows that ICAM-1 reacted with anti-ICAM-1 but not anti-LDLR antibodies in a concentration-dependent manner. LDLR specifically and dose-dependently reacted with LDLR-specific but not ICAM-1-specific antibodies (Figure 2d). In subsequent experiments, the ICAM-1- and LDLR-specific antibodies were tested for their ability to block the binding of major and minor group RV strains to their receptors. The ELISA-based RV–receptor interaction assay was developed to allow investigation of components either specific for the receptor or specific for the virus regarding their capacity to inhibit the virus–receptor interaction. For this purpose, receptor proteins were coated on the ELISA plates and incubated with the RV strains. The binding of RV strains could be detected with RV strain-specific antibodies and an HRP-labeled secondary antibody followed by substrate and color reaction (Figure 1). In this setting, the plate-bound receptor can be pre-incubated with receptor-specific compounds or the virus can be pre-incubated with virus-specific agents (Figure 1, bottom). 

### 3.2. Major and Minor Group RV Strains Bind Specifically to Their Corresponding Receptor Proteins in the ELISA Assay

Figure 3a shows that major group RV-A as well as distantly related RV-B strains were able to bind specifically to the plate-bound ICAM-1. No binding was detected when plates were coated with the control protein HSA and when virus or virus and anti-RV antibodies were omitted (Figure 3a, left panel). No binding of minor group RV strains was detected to ICAM-1, confirming the specific interaction of major group RV strains to recombinant ICAM-1 (Figure 3b, left panel). However, minor group RV strains bound specifically to ELISA plate-bound LDLR but not to the control protein, HSA (Figure 3b, right panel). There was also no binding when virus or virus and RV-specific detection antibodies were omitted. Some weak binding of major group RV strains (i.e., RV8, RV18, RV54, RV89, RV14, and RV72) to LDLR was observed above signals obtained with the control protein HSA (Figure 3a, right panel). 

### 3.3. Polyclonal Antibodies Produced against ICAM-1 Strongly and Specifically Inhibit Major Group RV Binding to ICAM-1 

In addition to the commercial ICAM-1-specific antibodies, we produced an antiserum specific for ICAM-1 by immunizing a rabbit with recombinant ICAM-1. We tested this antiserum regarding its capacity to inhibit the binding of two distantly related major group RV strains (i.e., RV89 and RV14) to ICAM-1 (Figure 4). We found that the anti-ICAM-1 antibodies but not normal rabbit antibodies almost completely (i.e., >85% inhibition) blocked binding of RV89 and of RV14 to plate-bound ICAM-1 (Figure 4a,b, left panels). The rabbit anti-ICAM-1 antibodies were highly potent in blocking the RV-ICAM-1 interaction because they still reduced the binding of RV89 to ICAM-1 even after a dilution up to 1:100,000, whereas normal rabbit antibodies had no effect on RV–receptor binding (Appendix A). A very weak but discrete binding of RV89 and RV14 to LDLR was observed in the control experiment performed with plate-bound LDLR (Figure 4a,b, right panels). 

### 3.4. ELISA-Based RV–Receptor Interaction is More Sensitive in Showing Anti-ICAM-1 Effects than an Established Cell Culture-Based Virus Neutralization Assay

Next, we compared the ELISA-based RV–interaction assay with a traditional cell culture-based virus neutralization assay regarding the abilities of the assay to reveal the capacity of anti-ICAM-1 antibodies to inhibit the binding of RV89 to ICAM-1. We performed both analyses with the same virus preparation and antibody dilutions. In cell culture-based virus neutralization assay, HeLa cells were incubated for three hours with serial dilutions of anti-ICAM-1 antibodies before they were infected with 100 TCID_50_/cell of RV89. Anti-ICAM-1 antibodies specifically inhibited RV89 infection up to a dilution of the antibodies of 1:2500, whereas no inhibition of infection was observed with the normal rabbit antibodies at any dilution (Figure 5a,b). Using the ELISA-based RV–interaction assay, an inhibitory effect of anti-ICAM-1 antibodies on the virus receptor interaction was detected up to a dilution of 1:50,000 (Appendix A). Thus, the ELISA-based RV–interaction assay was 20-fold more sensitive in revealing the inhibitory effects of the anti-ICAM-1 antibodies. The differences in duplicate determinations in the ELISA were always less than 10%, but we noted rather large SDs in the cell culture-based virus neutralization assay (Figure 5). 

### 3.5. Inhibition of Major and Minor Group RV Binding to Their Receptors with Anti-ICAM-1 and Anti-LDLR Antibodies 

In the next set of experiments, we tested the commercial anti-ICAM-1 and anti-LDLR antibodies for their ability to inhibit the binding of major group RV89 and minor group RV2 to ICAM-1 and LDLR in the ELISA-based virus–receptor interaction assay (Figure 6). We found that at the given dilution, the commercial anti-ICAM-1 antibodies inhibited RV89 binding to ICAM-1 to a lower degree (i.e., approx. 63% inhibition) than the antibodies against recombinant ICAM-1 (i.e., >85% inhibition) (Figure 6a, Figure 4a). Anti-LDLR antibodies specifically inhibited minor group RV2 binding to LDLR, whereas anti-ICAM-1 antibodies had no inhibitory activity. However, at the used dilution, the inhibition by anti-LDLR antibodies was only approximately 21% (Figure 6b). Some very weak binding of RV89 to LDLR was observed, which appeared reduced upon pre-incubation of LDLR with anti-LDLR antibodies (Figure 6a, right panel). No binding of minor group RV2 to ICAM-1 was observed (Figure 6b, left panel). 

### 3.6. Soluble Recombinant Receptors Specifically Inhibit Major and Minor RV Binding to Their Receptors

We then investigated if soluble recombinant receptors can be used to block major and minor RV binding to the immobilized receptors and what concentration of soluble receptors may be needed for inhibition. The left panel of Figure 7 shows that pre-incubation of RV89 with soluble ICAM-1 almost completely inhibited the binding of the virus to ICAM-1. The result of more than 95% inhibition of RV89 binding to ICAM-1 was achieved with 50 µg/mL ICAM-1 and a concentration of 25 µg/mL of purified RV89. A greater than 60% inhibition of minor group RV2 to LDLR was observed when the virus was pre-incubated with soluble LDLR (Figure 7, right panel). The inhibition of RV2 binding to LDLR was obtained with 5 µg/mL RV2 in the presence of 50 µg/mL LDLR. The specificity of the assays was ensured by several controls, including pre-incubation omission of virus and receptor, omission of virus or omission of receptor, and detection antibodies (Figure 7). 

### 3.7. ELISA-Based RV–Receptor Interaction Assay Allows Identifying Virus-Neutralizing Antibodies

Finally, we studied if the ELISA-based RV–receptor interaction assay could be used to screen RV-specific antibodies for their ability to block the RV–receptor interaction. We therefore pre-incubated RV89 with rabbit antibodies that had been produced against the RV89 coat proteins VP1 [27], VP2, VP3, or VP4. For control purposes, we performed pre-incubation only with buffer containing 0.5% BSA. Interestingly, we found that anti-VP1 and anti-VP2, but not anti-VP3 or anti-VP4 antibodies, inhibited the binding of RV89 to ICAM-1 (Figure 8). The inhibition produced with anti-VP1 and anti-VP2 antibodies was very strong (i.e., approximately 87%), indicating that the antisera contain virus-neutralizing antibodies. The anti-VP1 antiserum was already reported to exhibit virus-neutralizing activity in earlier cell culture-based virus neutralization assays [27,28]. 

## 4. Discussion

In this study, we established a simple and robust ELISA-based assay that can be used to investigate the interaction between major and minor group RVs with their corresponding receptors, ICAM-1 and LDLR. RV infections are the major cause of recurrent common colds and one of the most important trigger factors exacerbating respiratory illnesses such as wheeze, asthma, and chronic obstructive pulmonary disease (COPD). Major challenges to the development of RV vaccines and compounds (e.g., therapeutic antibodies, receptor derivatives, and molecular compounds) that inhibit RV infections are the diversity of RV types and their targeting of different receptors. For the development of RV vaccines, assays are required that allow for studying large numbers and different concentrations of antisera and antibodies induced by the vaccine candidate molecules in experimental animal models and human subjects regarding their ability to inhibit the binding of different RV species to their receptors. We demonstrated that the developed ELISA assay allows for the analysis of the binding of several major and minor RVs, including various RV-A and RV-B species to ICAM-1 and LDLR, respectively. Our assay could, in principle, be also extended to RV-C species because the corresponding receptor protein, cadherin-related family member 3 (CDHR3) protein, is known, and the binding sites have been mapped [29]. However, as RV-C isolates are difficult to obtain and were not available to us, we could not include RV-C in our studies. The major difference between our assay and traditional cell culture-based virus neutralization assays is that our assay demonstrates the direct binding of the virus to the receptor proteins without the need for cells. This is a major advantage because antisera produced in animals by immunization or human sera and tissue fluids may contain factors such as complements, which can be toxic to cells or may even have the opposite effects by enhancing cell growth [20,30,31]. When we compared our ELISA-based RV–receptor interaction assay with a traditional cell culture-based virus neutralization assay for inhibition of RV binding to ICAM-1 by anti-ICAM-1 antibodies, we found that it was 20-fold more sensitive and the results for higher dilutions of the antibodies appeared to be more consistent. Performing an ELISA assay is much faster and less arduous because no multiple-day cultivation of cells is needed. Cell culture-based assays are not suitable for large-scale screening of compounds (e.g., antibodies and antisera induced by different vaccine candidates) and different dilutions thereof. In contrast, the ELISA-based interaction assay should be a suitable platform for large-scale compound screening because it can be performed with plate-coated receptor proteins and virus binding can be quantified via a semi-quantitative colorimetric reaction by ELISA. We used labeled virus-specific antibodies for the detection of receptor-bound virus, but our assay may be further improved by using directly labeled virus to expand its use and application for drug discovery.

A possible limitation of the ELISA-based interaction assay is that it is focused on substances, which directly interfere with the virus–receptor interaction, and thus, cannot be used to test for the protective effects of antibodies that are not due to direct inhibition of the virus–receptor interaction. For example, an antibody was described that may protect against RV by inducing uncoating of RV [32]. Likewise, the virus–receptor interaction assay will not be suitable for studying RV-specific cellular immunity or certain indirect antibody-dependent effects on RV, such as antibody-dependent cellular cytotoxicity, complement activation, or opsonization. However, the latter effects also cannot be studied in cell culture-based assays but may need sophisticated in vivo models [21]. 

In our study, we demonstrated the usefulness of the ELISA-based RV–receptor interaction assay for several RV-specific treatment strategies. In this first series of experiments, we demonstrated that binding of major group RVs to ICAM-1 can be blocked with anti-ICAM-1 antibodies, and binding of minor group RVs to LDLR was inhibited by anti-LDLR antibodies. Even before the elucidation of the structure of RV in the complex with ICAM-1 [33], specific antibodies were developed with the goal of using them for therapeutic approaches based on the inhibition of the RV–receptor interaction [34,35,36,37]. Then, an anti-human ICAM-1 antibody was demonstrated to inhibit RV-induced exacerbation of lung inflammation [38] in an experimental animal model. Within the context of performing the binding experiments with major and minor group RVs to their receptors, we made an observation for which we have no explanation and its potential relevance is unclear. We noticed a weak but above background binding of certain major group RVs to LDLR (Figure 3a) that was reduced by anti-LDLR antibodies (Figure 6a). This finding suggests that major group RVs may have some residual weak ability to interact with LDLR.

We further explored the utility of our interaction assay by testing if soluble receptor molecules, when pre-incubated with RV, may be able to block RV–receptor binding. Recombinant ICAM-1 and its two N-terminal domains were reported to block RV infectivity [39]; subsequently, ICAM-1 derivatives with similar properties were described [40,41]. The approach of using soluble ICAM-1 forms for preventing RV infections was then successfully tested in chimpanzees and in a clinical trial in human subjects [42,43]. 

In our RV–receptor interaction assay, we were also able to demonstrate that pre-incubation of major group RV with quite low concentrations of ICAM-1 (i.e., micrograms/mL) were sufficient to block RV binding to immobilized ICAM-1. Likewise, we showed that pre-incubation of minor group RV with soluble LDLR inhibited minor group RV binding to its receptor. The latter result is in agreement with earlier observations that very low density lipoprotein receptor fragments shed from HeLa cells, as well as recombinant LDLR fragments inhibited minor group RV infection in vitro [44,45]. Similar as for ICAM-1, a conformational modification of the capsid was proposed to be the mechanism for the observed virus neutralization by LDLR [46]. However, our results demonstrated that the pre-incubation of RV with the soluble receptors strongly inhibits the binding of RV, indicating that blocking of the RV receptor binding site is an additional important mechanism through which soluble receptors inhibit RV infectivity.

Finally, for major group RV-A strain RV89, we demonstrated that polyclonal antisera produced against the coat proteins VP1 and VP2, but not against VP3 and VP4, inhibited the binding of the virus to ICAM-1. This experiment showed that the interaction assay can be used to test antibodies or antisera obtained by immunization with recombinant RV proteins, and suggests its usefulness for testing RV vaccine candidates and their ability to induce immunization antibodies, which block the RV–receptor interaction. The obtained results agree with earlier results obtained with virus escape mutants as well as with virus-specific monoclonal and polyclonal antibodies, indicating that VP1 and VP2 contain epitopes involved in the RV–receptor interaction [28,47,48,49,50,51,52,53,54]. However, we could not show that VP3- or VP4-specific antibodies could block the RV–receptor interaction, although results from virus escape mutants [47] and studies performed with specific antibodies [55,56,57,58] indicated that VP3 and VP4, respectively, may play a role in virus neutralization. This discrepancy may be explained by VP3 and VP4 not being directly involved in receptor binding but may contribute to virus neutralization via other mechanisms. 

## 5. Conclusions

We established a rapid, simple, and robust ELISA assay for studying the interaction between major group and minor group RV strains with their receptors. This assay should be useful for the screening of sera from RV-infected patients, therapeutic antibodies or antibody derivatives, candidate molecules for RV vaccines regarding their potential to induce blocking antibodies, receptor derivatives or mimics, as well as chemical compounds to pave the road toward RV-specific therapeutic and preventive strategies and should facilitate the preclinical development of RV vaccines.

## Figures and Tables

**Figure 1 vaccines-08-00315-f001:**
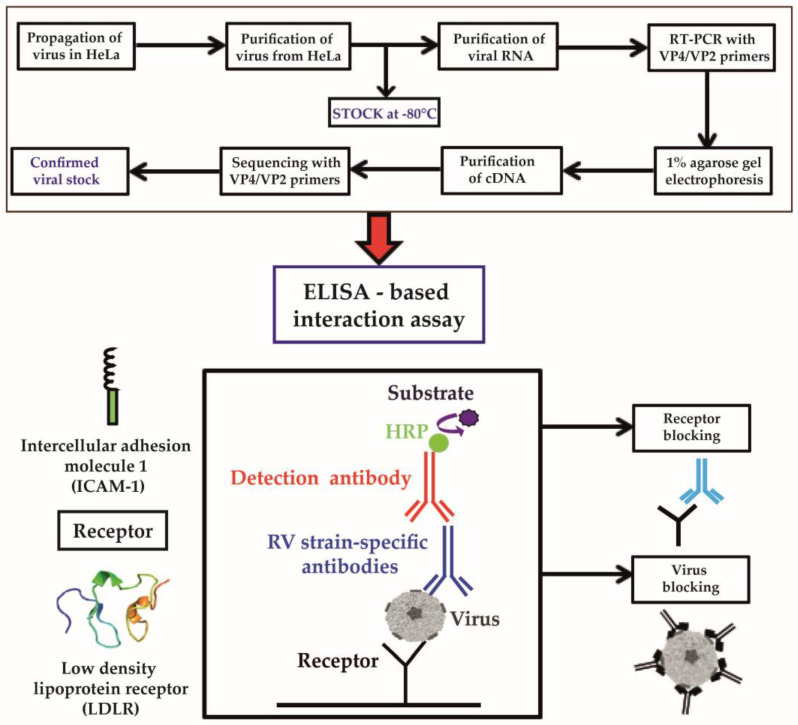
Flow chart summarizing the experimental design and reagents for the ELISA-based virus–receptor interaction assay. (top) The preparation of viral strains and the confirmation of their identity by reverse transcription (RT)-PCR. (bottom) The scheme for the ELISA detection of virus–receptor interactions.

**Figure 2 vaccines-08-00315-f002:**
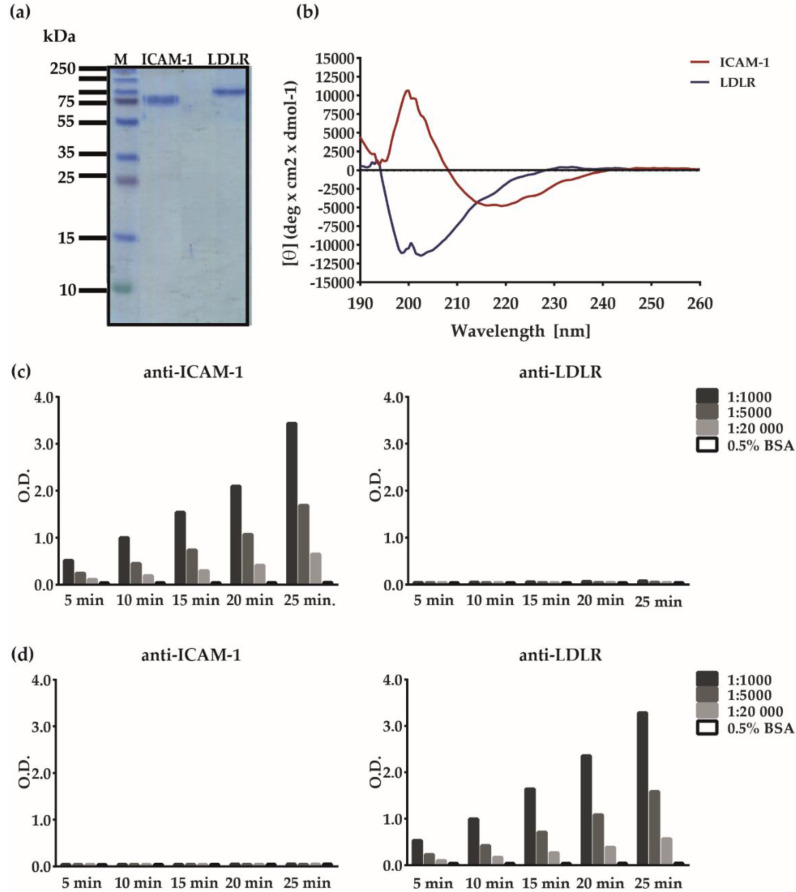
Characterization of recombinant human ICAM-1 and LDLR. (**a**) Coomassie-blue stained SDS-PAGE with recombinant human ICAM-1 (lane 1) and LDLR (lane 2). A molecular-weight marker (kDa) is indicated on the left. (**b**) Mean residue ellipticities (θ, *y*-axis) of ICAM-1 (red) and LDLR (blue) recorded at different wavelengths (nm, *x*-axis) by circular dichroism (CD) spectroscopy. Detection of (**c**) ICAM-1 or (**d**) LDLR with serial dilutions of anti-ICAM-1 antibodies (c = 1 µg/mL; left), anti-LDLR antibodies (c = 0.2 µg/mL; right) or bovine serum albumin (BSA)-containing buffer alone (BSA) by ELISA. Shown are means of optical density (O.D.) values corresponding to bound antibodies (y-axes) measured at different time points (x-axes). The variations of individual duplicate results of ELISAs were less than 5%. The antibody titration in (**c**) and (**d**) was done once.

**Figure 3 vaccines-08-00315-f003:**
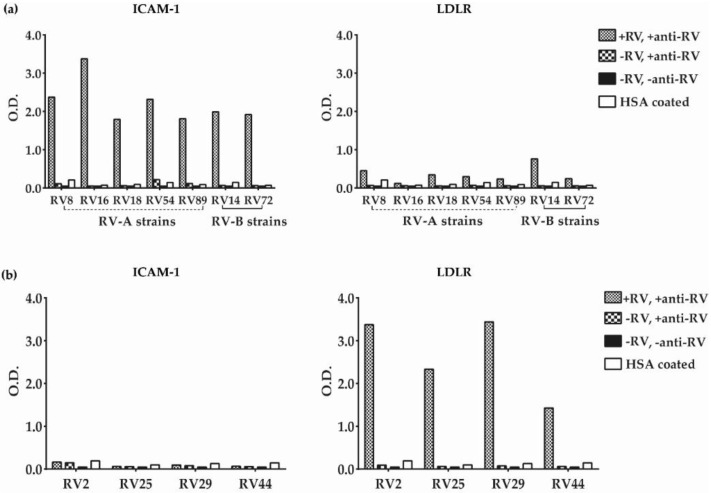
Binding of major group and minor group RVs to ICAM-1and LDLR. Shown are O.D. values (y-axes) corresponding to receptor (ICAM-1 or LDLR)-bound (**a**) major or (**b**) minor group RVs (x-axes) as determined by ELISA. Tested major group RV-A or RV-B species (**a**) and minor group RV-A species (**b**) are indicated on the x-axes. The analyses were performed with virus (50 µg/mL RV) and, for control purposes, without virus (−RV), without virus and detection antibodies (−RV, −anti-RV) or with HSA- instead of receptor-coated plates (HSA coated). The results are means of duplicate determinations with a variation of less than 5%. RV–receptor binding experiments were repeated two times.

**Figure 4 vaccines-08-00315-f004:**
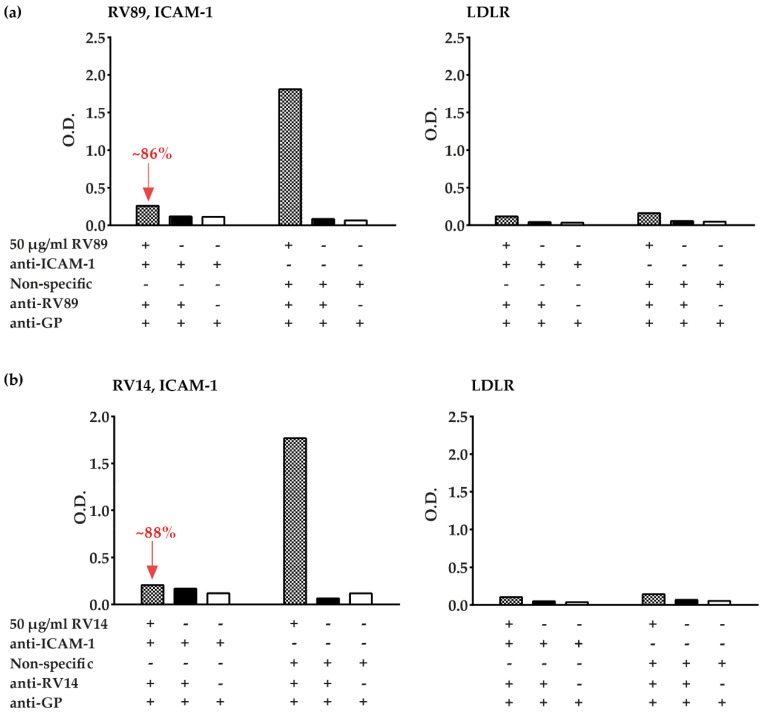
Inhibition of RV89 and RV14 binding to plate-bound receptors by receptor-specific antibodies. Binding of RV89 (upper part) (**a**) and RV14 (lower part) (**b**) to plate-bound ICAM-1 (left panels) or plate-bound LDLR (right panels) is reported as mean O.D. values (y-axes) in the presence (+) or absence (−) of virus, anti-ICAM-1 antibodies, non-specific antibodies, anti-RV detection antibodies, and secondary detection antibodies (anti-GP) (below x-axes). The percentage inhibitions of RV89 binding and RV14 binding obtained with anti-ICAM-1 antibodies versus non-specific antibodies are indicated in red. The results are means of duplicate determinations with a variation of less than 10%. Inhibition experiments were repeated three times.

**Figure 5 vaccines-08-00315-f005:**
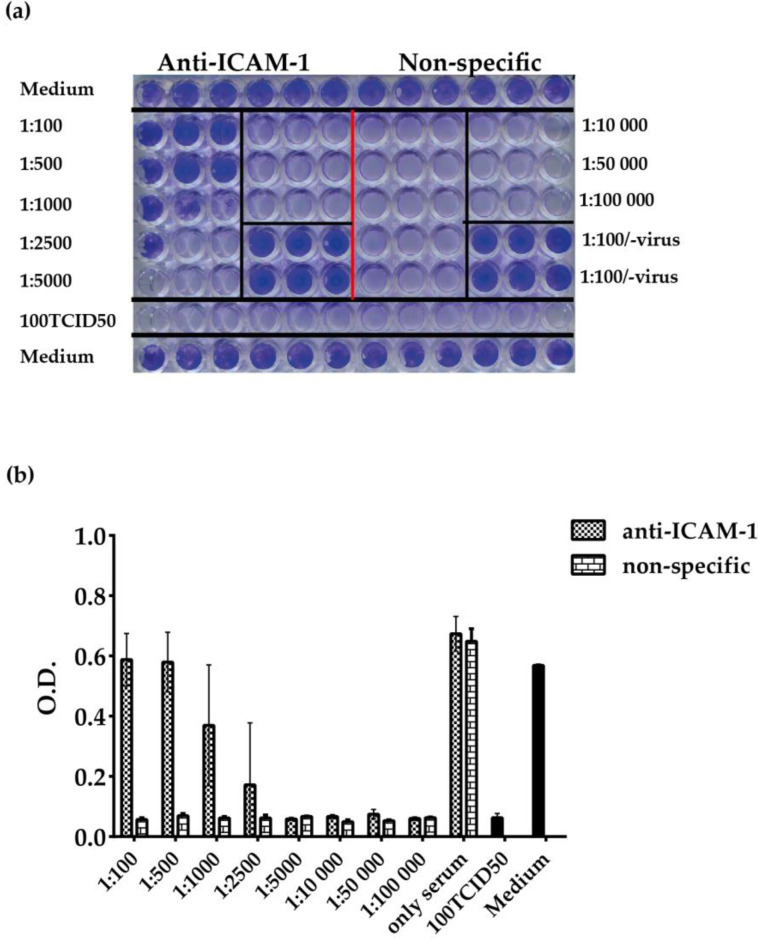
Inhibition of RV89 infection by anti-ICAM-1 antibodies determined by cell culture-based virus inhibition assay. (**a**) Crystal violet-stained viable cells incubated only with anti-ICAM-1 or non-specific antibodies (top), different dilutions of antibodies (1:100–1:100,000); anti-ICAM-1 and non-specific antibodies separated by red vertical line) in the presence of virus (100 TCID_50_), 1:100 diluted antibodies without virus, cell incubated only with virus (100 TCID_50_), or only with medium (bottom). (**b**) Mean O.D. values corresponding to viable cells in triplicate wells ±SDs (*y*-axis) of the above experiment (*x*-axis: conditions applied). Neutralization tests were repeated two times.

**Figure 6 vaccines-08-00315-f006:**
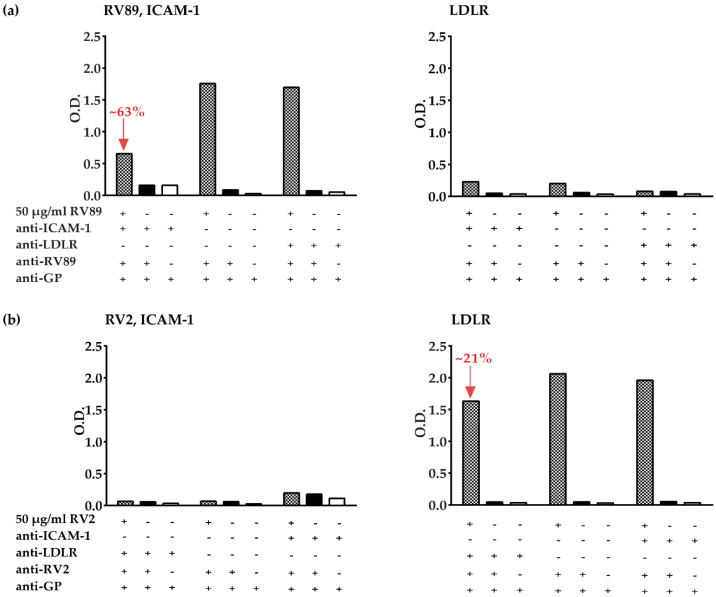
Inhibition of major group RV89 and minor group RV2 binding to plate-bound receptors by receptor-specific antibodies. Binding of RV89 (upper part) (**a**) and RV2 (lower part) (**b**) to plate-bound ICAM-1 (left panels) or plate-bound LDLR (right panels) reported as mean O.D. values (y-axes) in the presence (+) or absence (−) of virus, commercial anti-ICAM-1 antibodies, commercial anti-LDLR antibodies, anti-RV detection antibodies, and secondary detection antibodies (anti-GP) (see below x-axes). The percentage inhibitions of RV89 binding and RV2 binding obtained with commercial anti-receptor antibodies compared to without anti-receptor antibodies are indicated in red. The results are means of duplicate determinations with a variation of less than 10%. Inhibition experiments were performed once.

**Figure 7 vaccines-08-00315-f007:**
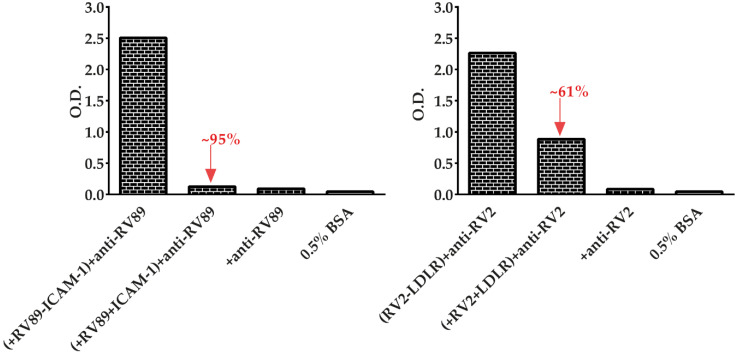
Auto-inhibition of the binding of major group RV89 and minor group RV2 to their receptors. Binding of RV89 to ICAM-1 (**left**) and of RV2 to LDLR (**right**) without (first column) or with (second column) pre-incubation with the receptor plus controls in which virus and receptor (third column) or virus, receptor and detection antibodies (fourth column) were omitted, reported as mean optical density (i.e., O.D.) values (y-axes). The percentage inhibitions of RV89 and RV2 binding produced by pre-incubation with receptors are indicated in red. The results are means of duplicate determinations with a variation of less than 10%. Inhibition experiments with soluble receptors were performed two times.

**Figure 8 vaccines-08-00315-f008:**
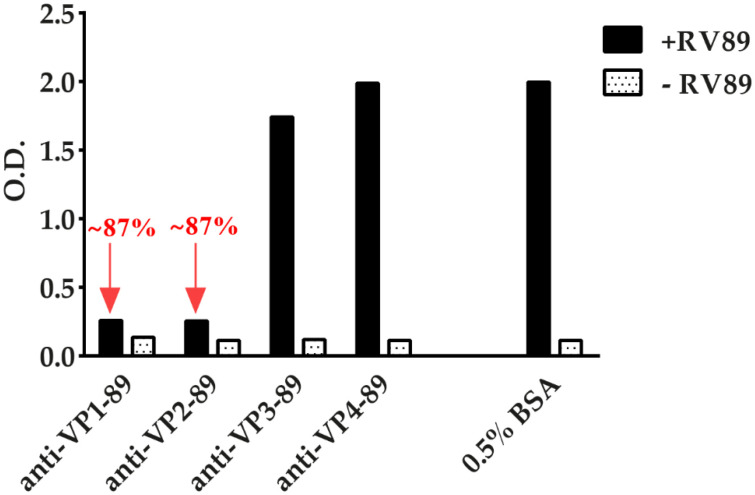
Inhibition of RV89 binding to ICAM-1 by antibodies specific for each of the four viral capsid proteins (VP). Binding of RV89 to ICAM-1 after pre-incubation of virus with antibodies against VP1, VP2, VP3, and VP4 compared to pre-incubation with buffer containing BSA only reported as O.D. (y-axes). The inhibitions of RV89 binding by anti-VP1 and anti-VP2 antibodies are indicated by red arrows. Omission of RV89 served as a negative control. The results are means of duplicate determinations with a variation of less than 10%. Inhibition experiments were performed three times.

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
