# Peer review of "ELISA-Based Assay for Studying Major and Minor Group Rhinovirus–Receptor Interactions"

_vaccines, 2020, doi:10.3390/vaccines8020315_

Round 1

Reviewer 1 Report

This manuscript describes the development of an ELISA-based testing platform for studying major and minor group rhinovirus-receptor interactions. This ELISA based testing assay was used to screen receptor-specific and RV-specific antibodies. The major advance of this work enables receptor-based discovery of vaccines with high sensitivity as compare to cell culture-based virus neutralization test. Although, this system is not suitable for high-throughput screening and analysis, it provides the most comprehensive platform for the screening of vaccines, antibodies and anti-viral drugs that has been described to date.

Notably, it describes assays that measure virus-receptor interactions with use of primary and secondary detection antibody which limits assays to be run at lower conversion with non-specific interactions of RV strain-specific antibodies. In addition, this leads to the limitations with reproducibility of data with respective to the negative control. Overall, this is a very solid effort that describes useful tools but has shortcomings in terms of utilization and application of this methodology for HTS to discover novel vaccine or anti-viral drug candidates with this tool.

  • For figure 2c which shows ICAM-1 reactivity in concentration dependent manner. Authors should provide concentrations used in this figure separately. Similarly, for Figure 2d as well. Please add legends to X-axis.
  • For Figure 2c and 2d, what is the reliability of the results? Does this experiment was performed only once if not then what is the standard deviation? Moreover, almost all the data provided in this manuscript requires similar explanation. Authors should be very careful with the reproducibility of the data, it must be replicated before addressing final conclusion.
  • All data set should be modified with inclusion of standard deviation.

Author Response

This manuscript describes the development of an ELISA-based testing platform for studying major and minor group rhinovirus-receptor interactions. This ELISA based testing assay was used to screen receptor-specific and RV-specific antibodies. The major advance of this work enables receptor-based discovery of vaccines with high sensitivity as compare to cell culture-based virus neutralization test. Although, this system is not suitable for high-throughput screening and analysis, it provides the most comprehensive platform for the screening of vaccines, antibodies and anti-viral drugs that has been described to date.

Notably, it describes assays that measure virus-receptor interactions with use of primary and secondary detection antibody which limits assays to be run at lower conversion with non-specific interactions of RV strain-specific antibodies. In addition, this leads to the limitations with reproducibility of data with respective to the negative control. Overall, this is a very solid effort that describes useful tools but has shortcomings in terms of utilization and application of this methodology for HTS to discover novel vaccine or anti-viral drug candidates with this tool.

 REPLY: We thank the reviewer for the kind evaluation of our study. We agree with the reviewer that the use of primary and secondary detection antibodies might be a limitation for these assays but the results shown in our study are already very encouraging. We have included in the discussion a statement that the assay maybe improved by using directly labelled virus and that this may expand the utilization and application of our assay for drug discovery (see lines 526-528 of the revised manuscript).   

For figure 2c which shows ICAM-1 reactivity in concentration dependent manner. Authors should provide concentrations used in this figure separately. Similarly, for Figure 2d as well. Please add legends to X-axis.

REPLY: Following the reviewer’s suggestion, we have added to the legend of Figure 2 information regarding the concentrations of the antibody stocks used in these experiments (see revised legend for Figure 2, line 322). 

For Figure 2c and 2d, what is the reliability of the results? Does this experiment was performed only once if not then what is the standard deviation? Moreover, almost all the data provided in this manuscript requires similar explanation. Authors should be very careful with the reproducibility of the data, it must be replicated before addressing final conclusion.

REPLY: We agree with the reviewer that reproducibility and variation of data is a very important issue. The ELISA experiments in Figure 2c and 2d were obtained as duplicate determination with less than 5% of variation of the individual measurements. The variation of duplicates for all other determinations was less than 10%. We have indicated this information in the revised manuscript (see lines 215-216 of the revised manuscript, lines 324-325).

All data set should be modified with inclusion of standard deviation.

REPLY: As stated above all results are presented as means of duplicates with less than 5%-max. 10% variation as indicated for each of the experiments. It is therefore not possible to calculate standard deviations but we have indicated in the legends to the individual figures the low variation between the duplicates (see lines 215-216 of the revised manuscript, revised figure legends lines 324-325, 361-362, 385-386, 451, 471-472, 493-494, 408, 605-606).

Reviewer 2 Report

In the manuscript "ELISA-based assay for studying major and minor group rhinovirus-receptor interactions” Pazderova P and colleagues established and characterized an ELISA-based interaction assay utilizing purified RVs and the corresponding human receptor proteins suggesting that as useful for the preclinical evaluation for preventive and therapeutic strategies targeting the RV-receptor interaction such as vaccines, antibodies and anti-viral compounds.

The main goal was to build an ELISA-based assay to investigate the interaction of intact RV strains with their corresponding receptors. To do that they the interaction assay was developed to allow investigate components both specific for the receptor or specific for the virus regarding their capacity to inhibit the virus binding.

All the figures lack the statistics meaning significance and errors bars (except one), please provide and indicate also the statistical information (meaning duplicates, experimental repetition, which test they used, etc.) in the captions.

Did the authors considered to use fluorescent antibodies instead ABTS? They could obtain a bigger dynamic range, that is interesting.

Line 31-32 “The assay turned out to be more sensitive than a cell culture-based virus neutralization test”, I wonder how the authors can state this. Given that several papers demonstrate the neutralizing effect of compounds and/or antibodies, this statement should be accompanied by the standard observation (meaning neutralizing test) to corroborate the data. Authors may want report previous or supplemental results (i.e., a table) showing the neutralizing effect of the antibodies they tested.

Line 651: there are overlapping characters

The MS is very well written, simple and clear in the narrative and I really appreciate they  discussed all the limitations of the study, which would highlight a strong analytical sense and scientific responsibility, meaning that they got the weaknesses of the study and its design.

Author Response

In the manuscript "ELISA-based assay for studying major and minor group rhinovirus-receptor interactions” Pazderova P and colleagues established and characterized an ELISA-based interaction assay utilizing purified RVs and the corresponding human receptor proteins suggesting that as useful for the preclinical evaluation for preventive and therapeutic strategies targeting the RV-receptor interaction such as vaccines, antibodies and anti-viral compounds.

The main goal was to build an ELISA-based assay to investigate the interaction of intact RV strains with their corresponding receptors. To do that they the interaction assay was developed to allow investigate components both specific for the receptor or specific for the virus regarding their capacity to inhibit the virus binding.

All the figures lack the statistics meaning significance and errors bars (except one), please provide and indicate also the statistical information (meaning duplicates, experimental repetition, which test they used, etc.) in the captions.

REPLY: We have performed all measurements in duplicates on the same plate with variations of results of less than 10%. This has been indicated in the revised manuscript and the Figure legends (see lines 215-216 of the revised manuscript, revised figure legends lines 324-325, 361-362, 385-386, 451, 471-472, 493-494, 408, 605-606).

Did the authors considered to use fluorescent antibodies instead ABTS? They could obtain a bigger dynamic range that is interesting.

REPLY: We thank the reviewer for this suggestion. In fact we have used ABTS in our experiments which gives a very good dynamic range of two orders of magnitude.

Line 31-32 “The assay turned out to be more sensitive than a cell culture-based virus neutralization test”, I wonder how the authors can state this. Given that several papers demonstrate the neutralizing effect of compounds and/or antibodies, this statement should be accompanied by the standard observation (meaning neutralizing test) to corroborate the data. Authors may want report previous or supplemental results (i.e., a table) showing the neutralizing effect of the antibodies they tested.

REPLY: In fact our statement was already corroborated by the direct comparison of the virus neutralization assay in Figure 5 with our assay shown in Figure S2. The results of this direct comparison are described in the revised manuscript (see lines 391-403).

Line 651: there are overlapping characters

REPLY: It seems that the overlap was due to conversion of the manuscript into a PDF. We have checked in the revised manuscript and it looked ok.

The MS is very well written, simple and clear in the narrative and I really appreciate they discussed all the limitations of the study, which would highlight a strong analytical sense and scientific responsibility, meaning that they got the weaknesses of the study and its design.

REPLY: We thank the reviewer for the positive comments regarding our work.